# Prolonged Infusion of β-Lactams Decreases Mortality in Patients with Septic Shock: A Retrospective before-and-after Study

**DOI:** 10.3390/antibiotics10060687

**Published:** 2021-06-08

**Authors:** Daniel Christoph Richter, Maximilian Dietrich, Lazar Detelinov Lalev, Felix C. F. Schmitt, Mascha Onida Fiedler, Thomas Bruckner, Dominic Stoerzinger, Ute Chiriac, Sabrina Klein, Thilo Hackert, Thorsten Brenner, Alexander Brinkmann, Markus A. Weigand

**Affiliations:** 1Department of Anesthesiology, Heidelberg University Hospital, Im Neuenheimer Feld 420, 69120 Heidelberg, Germany; lazardetelinov.lalev@med.uni-heidelberg.de (L.D.L.); felix.schmitt@med.uni-heidelberg.de (F.C.F.S.); mascha.fiedler@med.uni-heidelberg.de (M.O.F.); markus.weigand@med.uni-heidelberg.de (M.A.W.); 2Institute of Medical Biometry & Biostatistics, Im Neuenheimer Feld 130.3, 69120 Heidelberg, Germany; bruckner@imbi.uni-heidelberg.de; 3Department of Pharmacy, Heidelberg University Hospital, Im Neuenheimer Feld 670, 69120 Heidelberg, Germany; dominic.stoerzinger@med.uni-heidelberg.de (D.S.); ute.chiriac@med.uni-heidelberg.de (U.C.); 4Department of Infectious Diseases, Medical Microbiology and Hygiene, Im Neuenheimer Feld 324, 69120 Heidelberg, Germany; sabrina.klein@med.uni-heidelberg.de; 5Department of General, Visceral and Transplantation Surgery, Heidelberg University Hospital, Im Neuenheimer Feld 420, 69120 Heidelberg, Germany; thilo.hackert@med.uni-heidelberg.de; 6Department of Anesthesiology and Intensive Care Medicine, University Hospital Essen, University Duisburg-Essen, Hufelandstraße 55, 45147 Essen, Germany; thorsten.brenner@uk-essen.de; 7Department of Anesthesiology, Heidenheim Hospital, Schloßhaustraße 100, 89522 Heidenheim, Germany; alexander.brinkmann@kliniken-heidenheim.de

**Keywords:** prolonged infusion, β-lactams, septic shock, mortality, antibiotic therapy, critical care

## Abstract

Septic shock substantially alters the pharmacokinetic properties of β-lactams with a subsequently high risk of insufficiently low serum concentrations and treatment failure. Considering their pharmacokinetic (PK)/pharmacodynamic (PD) index, prolonged infusions (PI) of β-lactams extend the time that the unbound fraction of the drug remains above the minimal inhibitory concentration MIC (*f*t _>MIC_) and may improve patient survival. The present study is a monocentric, retrospective before-and-after analysis of septic shock patients treated with β-lactams. Patients of the years 2015–2017 received intermittent bolus application whereas patients of 2017–2020 received PI of β-lactams. The primary outcome was mortality at day 30 and 90 after diagnosis of septic shock. Mortality rates in the PI group were significantly lower on day 30 (PI: 41%, *n* = 119/290 vs. IB: 54.8%, *n* = 68/114; *p* = 0.0097) and day 90 (PI: 47.9%, *n* = 139/290 vs. IB: 62.9%, *n* = 78/124; *p* = 0.005). After propensity-score matching, 30- and 90-day mortality remained lower for the PI group (−10%, *p* = 0.14). PI was further associated with a reduction in the duration of invasive ventilation and a stronger decrease in SOFA scores within a 14 day-observation period. PI of β-lactams was associated with a significant reduction of mortality in patients with septic shock and may have beneficial effects on invasive ventilation and recovery from sepsis-related organ failure.

## 1. Introduction

Sepsis has emerged as one of the most challenging global health problems, with increasing patient numbers (49 million cases) and persistently high mortality rates (11 million) [1,2]. The German INSEP study [3] included 1503 patients with sepsis and septic shock and found an overall intensive care unit (ICU) mortality of 34.3% (compared with 6% in patients without sepsis) and a 40.4% hospital mortality of septic patients. 

Rapid administration of broad-spectrum antibiotics is regarded a key feature of sepsis treatment [4]. Prospective studies found a 2 to 8% increase in mortality with each hour of antibiotic delay [5,6]. β-lactams like piperacillin/tazobactam (TZB) and carbapenems (CP) are considered frontline broad-spectrum antibiotics and are the most frequently prescribed substances for the treatment of sepsis and septic shock [7,8]. Sepsis and septic shock induce pathophysiological changes that gravely alter the pharmacokinetic properties of the hydrophilic β-lactams [9,10,11] and ultimately may culminate in low β-lactam serum concentrations [12,13,14,15] jeopardizing treatment success [16,17]. To provide critically ill patients with effective antimicrobial therapy, guidelines strongly support efforts to optimize antimicrobial dosing [4,18,19]. As time-dependent antibiotics, antimicrobial potency of β-lactams is predominantly determined by the time (T) that the unbound fraction (*f*) of the drug remains above the pathogen-specific minimal inhibitory concentration (MIC) [20,21]. However, in most immediate clinical scenarios, causative pathogens, MIC and measured serum concentrations of β-lactams are not readily available for treating physicians. In this case, pragmatic PK-targets considering epidemiological cut-off values (ECOFF) have been proposed (50–100%*f*T_>2-4MIC_ of possible resistant targets, like *Pseudomonas aeruginosa*) to ensure the best possible antimicrobial exposure. The prospective DALI study by Roberts et al. [22] investigated antimicrobial exposure in a cohort of critically ill patients treated with standard doses of eight distinct β-lactams (including CP and TZB). 67% of patients were treated with intermittent bolus (IB) application while 33% received a prolonged infusion (PI). The investigators found acceptable PK-target attainment for the most conservative targets (Meropenem, 50% *f*T_>MIC_: 95%; TZB, 50% *f*T_>MIC_: 80.6% of patients) but poor target attainment for pragmatic PK-targets aimed for in sepsis and septic shock (100% *f*T_>4MIC_: 41.6% [Meropenem] and 30.3% [TZB]). Overall, 48.8% of patients achieved 50% *f*T_>4MIC_ and only 35% attained a target of 100% _f_T_>4MIC_. With regard to PK properties, PI are a promising compromise to IB. Applying an initial loading dose (LD) and infusing subsequent doses over a period of 2 to 4 h is a pragmatic way to effectively extend the *f*T_>MIC/ECOFF_ throughout the dosing interval. In contrast to continuous infusion (CI) of β-lactams, therapeutic drug monitoring (TDM) is not essentially required. Abdul-Aziz and co-workers performed a post-hoc analysis [23] of the DALI study [22] to investigate effects of PI on patient mortality and found significantly higher 30-day-survival rates for patients receiving PI for respiratory infections (PI 86.2% [*n* = 25/29] vs. IB 56.7% [17/30]) and patients with a SOFA score ≥9 (PI 73.3% [*n* = 11/15] vs. IB 35% [7/20]; *p* = 0.035). A recent meta-analysis by Rhodes et al. [24] that included 18 studies with 3401 patients treated with TZB concordantly reported a 1.77-fold higher odds of clinical cure (95% CI, 1.24–2.54) and a 1.22-fold higher odds of microbiological cure (95% CI, 0.84–1.77) in patients receiving PI. Older data by Falagas and co-workers [25] also reported lower mortalities for PI. Although, studies on PI of β-lactams comprise a broad spectrum of critically ill patients with varying degrees of disease severity and different foci of infection. For example, Abdul-Aziz et al. [23] included patients with a median SOFA score of 5 (IQR3-8) and predominantly respiratory (38.5%) and abdominal (27.5%) infections. We introduced PI of β-lactams in the surgical intensive care unit (ICU) at Heidelberg University Hospital in 2017. As we treat a relatively homogenous and severely ill cohort of patients with abdominal sepsis, the present analysis aims to investigate the effects of PI of β-lactams on mortality in severely ill patients with mainly abdominal septic shock. We matched the PI cohort to a comparable cohort of the years 2015–2017 when β-lactams were administered as IB.

## 2. Materials and Methods

### 2.1. Study Design

The present study is a monocentric, retrospective before-and-after analysis of electronically collected clinical data of the interdisciplinary surgical ICU at Heidelberg University Hospital, Germany. The local Institutional Ethics Committee approved the study (reference number: S-586/2020).We included adult patients (age >18 years) that were treated with β-lactam antibiotics for septic shock (according to the revised SSC definition [4]) between March 2015 and July 2020. Two groups were compared: patients of the years 2015–2017 received IB application, whereas patients of 2017–2020 received PI of β-lactams. 

### 2.2. Antimicrobial Treatment, Clinical Scores, Laboratory Data

Sepsis therapy was based on the recommendations of the SSC [4] and the German Sepsis Society (DSG) [26]. The local protocol for the prolonged β-lactam administration since April 2017 consisted of an initial LD (30-min bolus infusion) and PI of consecutive doses over a period of 3 h using a rate-controlled syringe pump. Before April 2017 β-lactam antibiotics were administered as an intermittent bolus of a 20-min infusion. The LD comprised of one standard dose administered as bolus infusion. Despite the LD, patients in both groups (IB & PI) received standard doses of the respective β-lactams. Doses were corrected according to the manufacturers’ recommendation (mostly for acute kidney injury/renal failure). We did not conduct therapeutic drug monitoring (TDM).

The patients’ clinical and demographic data presented in this study were derived from the clinical documentation system. Vasopressor dependence and inotropic support were summarized and analyzed using the vasoactive-inotropic score (VIS) as proposed by Butts et al. [27]: VIS = dopamine dose (μg/kg/min) + dobutamine dose (μg/kg/min) + 100 × epinephrine dose (μg/kg/min) + 100 × norepinephrine dose (μg/kg/min) + 10,000 × vasopressin dose (U/kg/min) + 10 × milrinone dose (μg/kg/min)

Assessed laboratory data included leucocyte count, C-reactive protein (CRP), procalcitonin (PCT), creatinine, transaminases (aspartate and alanine aminotransferase), bilirubin and albumin levels. Sepsis-related organ failure assessment (SOFA) score [28] and acute physiology and chronic health evaluation II (APACHE 2) [29] scores were used to assess the degree of organ dysfunction/failure and as a surrogate for disease severity. Laboratory data, the amount of catecholamines administered were obtained at study inclusion, 48 h, 96 h, 168 h and 14 days after study inclusion.

### 2.3. Clinical Outcome Measures

We analyzed differences in mortality rates of patients treated with IB and PI. The primary outcome was mortality at day 30 and 90 after recognition of septic shock. In addition, overall ICU- and hospital mortality was assessed. Secondary outcomes included the ICU length of stay and the duration of invasive ventilation. Vasopressor dependence (VIS), the serum CRP, PCT and SOFA score were evaluated at enrollment 48 h, 96 h, 7 days and 14 days after study inclusion.

### 2.4. Statistical Methods

Data was collected with the aid of an electronic database system (Microsoft Excel^®^, Microsoft Deutschland GmbH, Unterschleißheim, Germany). The SAS System (Version 9.4, SAS, Cary, NC, USA) was used for statistical analyses. Descriptive statistics were done for the complete dataset. To adjust for baseline differences between the PI group and the IB group for the primary and secondary endpoint analysis, a propensity-score matching was performed including age, sex, SOFA- and vasoactive-inotropic score. For continuous variables and scores, mean, standard deviation (SD), minimum, median and maximum were calculated. Mean values with SD are presented in the manuscript. Absolute and relative frequencies of categorical variables are presented. The t-test test was used for the comparison of metric data between unpaired samples. For the comparison of categorical data, the chi-square test was used. Survival analysis was performed using a Kaplan-Meier graph with log-rank test and a fixed observation period of 30 days. Laboratory data and score courses over time were evaluated by analyses of covariance with repeated measurements. Appropriate statistical graphics were used to visualize the findings. 

## 3. Results

### 3.1. Baseline Characteristics

A total of 414 septic shock patients were eligible for inclusion into the final statistical analysis. The cohorts differed in sample size (IB group *n* = 124, PI group *n* = 290 patients). Both groups did not significantly differ regarding demographical baseline data (Table 1). Mean SOFA scores of 13.5 (3.3) (IB) and 12.6 (2.9) (PI)—although significantly different—however, indicated an equal severity of disease in both groups. Renal impairment measured as mean GFR was moderate (GFR IB 56 [46] vs. PI 56 [40] mL/min) (Table 1). 43% (*n* = 53/124) of patients in the IB group and 40% (*n* = 116/290) needed at least temporal renal replacement therapy for sepsis related acute renal failure. Abdominal foci were most common in both groups (IB: 66%, *n* = 82/124 and PI 76%, *n* = 221/290). For the primary and secondary endpoints, additional propensity-score matched analyses were performed. Both matched groups included 112 patients. The Propensity-Score matched PI group had a lower rate of known tumor disease. SOFA score was lower, APACHE II score was higher in the propensity-score matched PI group than in the propensity-score matched IB group. The baseline characteristics of all patients and the propensity-score matched cohorts are given in Table 1. Figure 1A depicts the distribution of antibiotics utilized for treatment. The doses per day and bodyweight of the most commonly used antibiotics are presented in the following:
**TZB:** PI group 0.2 g/kg/d (0.1)  vs. IB group 0.19 g/kg/d (0.06)**Meropenem:** PI group 0.05 g/kg/d (0.03)  vs. IB group 0.04 g/kg/d (0.01)**Imipenem:** PI group 0.04 g/kg/d (0.02)  vs. IB group 0.04 g/kg/d (0.01)

One hundred and thirty-four (134) patients (46.2%) of the PI group and 51 patients (41.1%) of the IB group had already received antibiotic therapy prior to the onset of septic shock. Overall, carbapenems (meropenem 60%, *n* = 249/414, imipenem 17%, *n* = 70/141) and piperacillin/tazobactam (22%, *n* = 89/414) were the most frequently prescribed antibiotics. Compared to the IB cohort, patients in the PI group were given TZB more frequently (12%, *n* = 15/124 vs. 26%, *n* = 74/290 (Figure 1B,C). 29.3% of patients of the PI group and 25.8% of patients of the IB group were additionally treated with antimycotics. 17.9% of patients of the PI group and 26.6% of patients of the IB group received antiviral medication. Consistent with the predominantly abdominal focus of infection, Gram-negative Enterobacterales and Gram-positive Enterococci were the leading pathogens (Table 2). In 36 of 414 patients (8.7%) septic shock was treated without evidence of a causative pathogen.

### 3.2. Primary Outcome

#### Prolonged β-Lactam Administration Was Associated with Lower Mortality

Unmatched analysis of the entire cohort suggested a significantly lower 30-day (−13.8%, *p* = 0.0097) and 90-day-mortality (−15%, *p* = 0.005) in patients receiving PI of the β-lactams. Survival estimates using a Kaplan-Meier-Plot of 30d-mortality (Figure 2) shows early partitioning of survival probability lines. The propensity-score matched analysis of the cohorts led to non-significant, however clinically still relevant, differences in 30 day-(−10.7%), 90 day-(−9.8%), ICU-(−9.9%), and hospital-mortality (−11.6%) (Table 3).

### 3.3. Secondary Outcome Parameters

#### 3.3.1. Prolonged β-Lactam Administration Was Associated with Shorter Duration of Invasive Ventilation

Patients treated with PI also had a shorter ICU stay in the unmatched (IB 15.8 d [19.4 d] vs. PI 13.6 d [16.1 d]; 0.219) and matched analysis (IB 17.1 d [19.9 d] vs. PI 13.5 d [15.7 d]; 0.139), the difference was not statistically significant in both analyses (Figure 3A). PI of β-lactams was associated with significantly shorter mean duration of invasive ventilation in both unmatched (IB 13.2 d [16.2 d] vs. PI 8.4 d [10.1 d]; *p* < 0.001) and propensity-score matched analysis (IB 14.2 d [16.6 d] vs. PI 8.3 d [10.6 d]; *p* = 0.002) (Figure 3B).

#### 3.3.2. PI of β-Lactams Was Associated with Faster Improvement of Organ Failure

Figure 4A–F depict the course of other parameters clinically used to predict responses to effective sepsis therapy. While the decline in laboratory infection surrogates (C-reactive protein [CRP] and procalcitonin [PCT]), APACHE II score and vasopressor dependence (VIS) did not significantly differ between both groups. PI of β-lactam antibiotics was associated with a stronger decrease of the SOFA score over a period of 14 days.

Number (*n*) of patients is presented below graph in red for the prolonged infusion and blue for the intermittent bolus group. SOFA Score: Sepsis-related organ failure assessment score. APACHE II: Acute Physiology And Chronic Health Evaluation II.

## 4. Discussion

This retrospective study investigated the effects of prolonged infused β-lactams on mortality of patients in septic shock in a German intensive care unit between 2015 and 2020. Critically ill patients with sepsis are prone to insufficiently low serum concentrations and consecutive treatment failure because of unique alterations of β-lactam PK [11,13,14,30,31,32]. PI of β-lactams evolved as a reaction to a better understanding of the PK/PD index of this group of antibiotics whereby prolongation of the fT_>MIC/ECOFF_ subsequently leads to a better antimicrobial exposure of the causative pathogens [33]. Previous investigations already suggested lower mortality rates and better clinical outcome using extended infusions in adults [23,24,25,34,35,36] and pediatric patients [37]. These effects are attributed to improved PK target attainment during extended infusions [36,38,39,40]. Critical patients included in the study of Roberts [22] and Abdul-Aziz et al. [23] had median SOFA scores of only 6 (IQR 3–9) and had varying sites of infection. Abdul-Aziz and co-workers [23] demonstrated significantly lower mortality for PI of β-lactams in patients with a SOFA score ≥9 and those with pneumonia. We analyzed a cohort of 414 severely ill patients with predominantly abdominal septic shock with mean SOFA scores of 13 predicting an a priori high mortality [41]. The monocentric nature of the study allowed for good comparability as depicted in the baseline characteristics given in Table 1. Contrary to Rhodes et al. [24], we exclusively evaluated only prolonged infusions of β-lactams. As a primary outcome measure, we found a significant reduction in mortality for the PI group. To control for interfering clinical properties, we performed a propensity-score matching and re-analyzed the primary outcome measure in the matched cohorts. While differences in mortality did not reach statistical significance, we still found markedly lower mortality rates when β-lactams were administered as a PI. Because of the unequal number of patients in both groups, generating pairs of patients as a prerequisite of matching was followed by a loss of about 50% of patients making it unlikely for a chi-square test to reach the significance level. Decreasing mortality rates by 10% in a cohort of patients with high probability of death—statistically significant or not—is, of utmost importance to patients and clinicians. Kaplan-Meyer estimates of survival show an early divergence of survival curves, suggesting a significant impact (*p* = 0.014) of PI on median survival probability. These findings strongly support the conclusion of a recently published review of Abdul-Aziz and colleagues whereby PI of β-lactams might be especially important for critically ill/immunocompromised patients and those who are more likely to have less susceptible Gram-negative infections [42]. Responding to the revised SSC guidelines [4], we changed our local sepsis protocol in 2017 which most likely explains that patients in the PI group received TZB considerably more often (26%) as between 2015–2017 (12%) while prescription patterns of meropenem did not change. As can be seen in Table 2, major changes in causative pathogens or an increase of infections caused by multidrug-resistant bacteria as a reason can be excluded. We further found a reduction of duration of invasive ventilation for patients in the PI group. We would like to argue that these effects are in line with prospective investigations highlighting the role of focus control and adequate antibiotic treatment [3,5,6,43,44]. On the other hand, uncontrolled or prolonged abdominal infection is accompanied by a gastrointestinal paralysis with subsequent distension of intestinal loops leading to a restrictive airway and higher incidence of micro-aspiration [45,46]. An interesting finding of our study is, that inflammatory markers like C-reactive protein and procalcitonin showed similar courses (Figure 4C,D) although PI was associated with lower mortality. An explanation is that CRP is an unreliable marker of infection in patients on surgical ICUs as CRP levels can vary as a consequence of surgical trauma or liver insufficiency. PCT is an inflammatory marker mainly well evaluated for (lower) respiratory tract infections and a general marker to distinguish bacterial from non-bacterial infections. While PCT is routinely used in abdominal infections its role in guiding therapy is still unclear [47].In what way the gut-lung axis and its role in immunomodulation contributes remains to be investigated [48]. Unlike previous prospective studies [38,39], we included patients on RRT or/and hepatic dysfunction. RRT may have unpredictable effects on serum concentrations of hydrophilic β-lactams and PK target attainment, according to recently published work [36,49]. We decided against a detailed evaluation of the subgroup of patients receiving RRT for two reasons. Firstly, serum concentrations were not available, and the study did not focus on PK effects. Secondly, we intended to analyze crude overall effect of a representative cohort of septic shock patients. With a fraction of 43% (IB) and 40% (PI) both cohorts show a rather high fraction of patients in need for temporal RRT. The high rates of RRT result from the inclusion criteria that focused on patients in septic shock. Further, baseline characteristics imply that the evaluated cohort constituted of older patients (IB: 65 y [13]; PI: 66 [13]) with high VIS. 

The study has some limitations. Data were drawn from a single center and comprised an exclusive cohort of patients with predominantly abdominal septic shock. Though our findings are in line with other prospective studies, findings cannot readily be generalized to other cohorts. Although we report microbiological data, resistance data and MICs were not available. Also, serum concentrations of β-lactams were not measured. Attaining PK targets is advised during treatment of critically ill patients [19]. Unfortunately, our data do not allow to link outcome (mortality) to specific PK-targets. Until 2020, we did not implement TDM-guided dose optimization in our department and adjusted β-lactam doses according to the manufacturers’ recommendation. Hence, we were not able to either investigate β-lactam toxicity [50,51,52,53] nor could we connect serum concentrations to outcome. Against this background, we are expecting the results of two interesting studies: The German TARGET study by Hagel et al. [54] (DRKS00011159) investigates the impact of TDM-guided dose adjustments in the ICU on survival of patients with sepsis and septic shock while the Australian BLING 3 study (NCT03213990) is still recruiting patients to evaluate clinical effects of continuously infused β-lactams versus intermittent bolus application.

## 5. Conclusions

The present study strongly endorses existing evidence of pivotal prospective studies and meta-analyses favoring PI of β-lactams in critically ill patients. PI was associated with a clinically relevant reduction of mortality in a severely ill subpopulation of ICU patients with septic shock and a shorter duration of ventilation. Further studies are necessary to investigate the effect of TDM-guided dose-optimization in addition to PI.

## Figures and Tables

**Figure 1 antibiotics-10-00687-f001:**
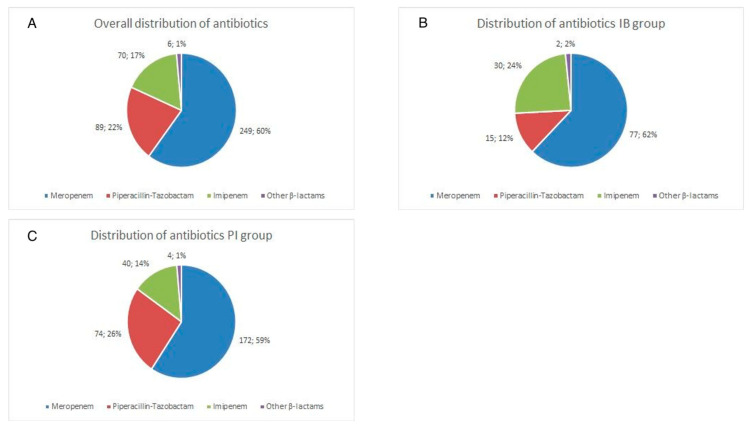
Distribution of Antibiotics in (**A**) Entire Cohort, (**B**) Intermittent bolus (IB) group and (**C**) prolonged infusion (PI) group.

**Figure 2 antibiotics-10-00687-f002:**
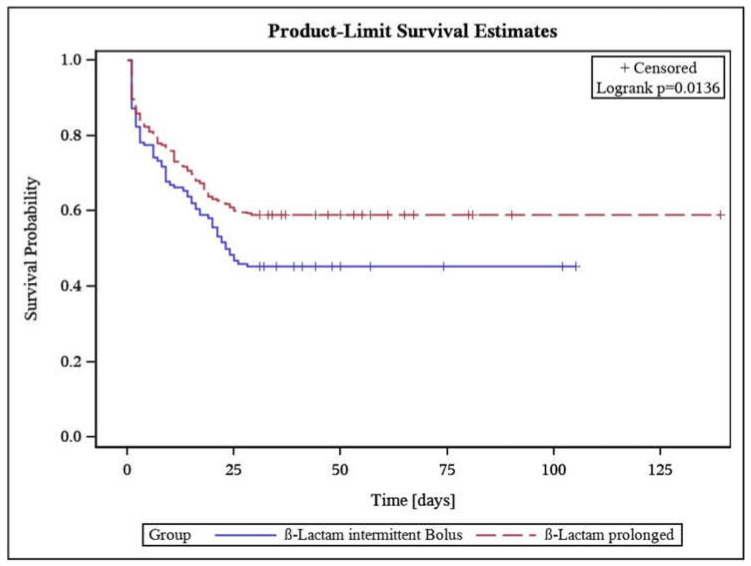
Kaplan-Meier product-limit survival estimates in intermittent bolus and prolonged infusion group.

**Figure 3 antibiotics-10-00687-f003:**
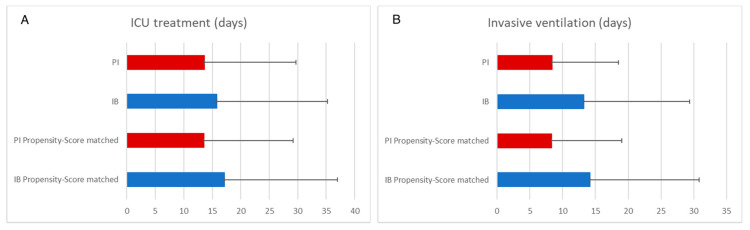
Secondary endpoints. Plot bars show mean values, error bars the standard deviation. PI: prolonged infusion. IB: intermittent bolus. (**A**) ICU treatment; (**B**) Invasive ventilation.

**Figure 4 antibiotics-10-00687-f004:**
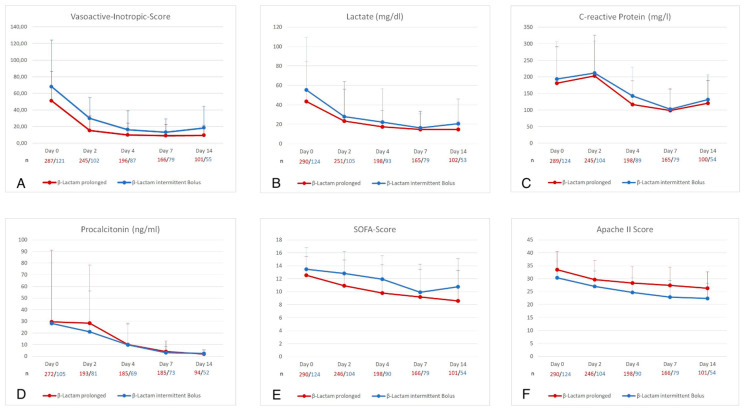
Course of Vasoactive-inotropic-score (**A**), Lactate (**B**), C-reactive protein (**C**), Procalcitonin (**D**), SOFA-Score (**E**), APACHE II Score (**F**) in baseline cohort over a 14 day observation period.

**Table 1 antibiotics-10-00687-t001:** Patient characteristics at study enrollment (onset of septic shock*) in baseline cohort and propensity-score analysis.

Baseline (*n* = 414)				Propensity-Score Matched (*n* = 224)		
Intermittent Bolus (*n* = 124)		Prolonged Infusion (*n* = 290)	*p*-Value	Intermittent Bolus (*n* = 112)	Prolonged Infusion (*n* = 112)	*p*-Value
Demographic data						
Male sex	86 (69%)	189 (65%)	0.4091	81 (72%)	81 (72%)	1.0000
Tumor disease	68 (55%)	150 (52%)	0.5610	63 (56%)	48 (43%)	0.0450
Organ transplant	16 (13%)	28 (10%)	0.4926	16 (14%)	7 (6%)	0.1323
Age	65 (13)	66 (13)	0.3792	64 (13)	64 (13)	0.9309
BMI [kg/m^2^]	29 (8)	29 (9)	0.9718	29 (8)	28 (10)	0.3679
Clinical condition						
SOFA score	13.5 (3.3)	12.6 (2.9)	0.0069	13.2 (3.0)	12.4 (3.1)	0.041
APACHE II Score GFR (Cockroft-Gault)	30.3 (6.4) 56 (46)	33.5 (7.1) 56 (40)	<0.0001 0.8994	29.9 (6.3) 60 (47)	33.7 (6.6) 57 (50)	<0.0001 0.6158
VIS [μg/kg/min]	68 (56)	51 (35)	0.0073	57 (37)	56 (40)	0.5616
Creatinine [mg/dL]	2.1 (1.2)	2.0 (1.2)	0.457	2.1 (1.2)	2.0 (1.3)	0.4988
Serum lactate [mg/dL]	55 (54)	44 (41)	0.0428	51 (49)	46 (47)	0.5728
Leukocytes [n/mL]	17 (12)	17 (12)	0.6721	17 (12)	15 (11)	0.5173
CRP [mg/L]	193 (113)	181 (182)	0.1036	194 (110)	193 (256)	0.1355
Procalcitonin [ng/mL]	28 (52)	30 (62)	0.0534	26 (45)	30 (50)	0.7274
SOT	% (*n*) 13 (16/124)	% (*n*) 10 (28/290)	0.5	% (*n*) 14 (16/112)	% (*n*) 6 (7/112)	0.1323
Liver	63 (10/16)	46 (13/28)		63 (10/16)	71 (5/7)	
Kidney	19 (3/16)	36 (10/28)		19 (3/16)	14 (1/7)	
Others	19 (3/16)	18 (5/28)		19 (3/16)	14 (1/7)	
Focus			0.04 *			0.108
Abdominal	66 (82/124)	76 (221/290)		65 (73/112)	78 (87/112)	
Pulmonary	17 (21/124)	11 (31/290)		18 (20/112)	10 (11/112)	
UTI	6 (7/124)	7 (20/290)		5 (6/112)	6 (7/112)	

Data are *n*(%) or mean (SD) unless otherwise specified. BMI: Body-mass-index. SOFA: Sequential Organ Failure Assessment score. APACHE II: Acute Physiology And Chronic Health Evaluation II. GFR: Glomerular filtration rate. VIS: Vasoactive-Inotropic score (μg/kg per min) = 10× milrinon dose (μg/kg/min) + 10,000× vasopressin dose (U/kg/min) + 100× (epinephrine dose [μg/kg/min] + norepinephrine dose [μg/kg/min]). CRP: C-reactive Protein. SOT: solid organ transplantation. UTI: Urinary tract infection. * administration of: 1 or more vasopressor to maintain a mean arterial pressure of 65 mmHg or higher + hydrocortisone + β-lactam antibiotic.

**Table 2 antibiotics-10-00687-t002:** Pathogen distribution.

		Intermittent Bolus (*n* = 179)		Prolonged Infusion (*n* = 451)	
		Lead Pathogen (*n* = 112)	Contributing Pathogen (*n* = 67)	Lead Pathogen (*n* = 265)	Contributing Pathogen (*n* = 186)
		% (n)	% (n)	% (n)	% (n)
**Gram-negative**	*E. coli*	25 (28)	20.9 (14)	29.8 (79)	16.1 (30)
	MDR—*E. coli* *	1.8 (2)	---	---	0.5 (1)
	*K. pneumoniae*	9 (10)	3 (2)	6.4 (17)	4.3 (8)
	MDR—*K. pneumoniae* *	none	1.5 (1)	0.4 (1)	---
	*K. oxytoca*	2.7 (3)	---	3.4 (9)	4.3 (8)
	*Klebsiella* spp.	---	---	0.4 (1)	1.1 (2)
	*Citrobacter freundii*	1.8 (2)	3 (2)	0.4 (1)	0.5 (1)
	*Citrobacter* spp.	1.8 (2)	---	---	1.1 (2)
	*Serratia marscescens*	2.7 (3)	---	0.8 (2)	0.5 (1)
	*Serratia* spp.	0.9 (1)	---	0.4 (1)	---
	*Proteus mirabilis*	2.7 (3)	1.5 (1)	4.5 (12)	1.6 (3)
	*Proteus* spp.	---	1.5 (1)	0.4 (1)	0.5 (1)
	*Enterobacter* spp.	3.6 (4)	---	2.3 (6)	5.4 (10)
	*Pseudomonas aeruginosa*	6.3 (7)	6 (4)	---	5.9 (11)
	MDR—*P. aeruginosa* ^#^	0.9 (1)	1.5 (1)	---	---
	*Bacteroides fragilis*	3.6 (4)	7.5 (5)	1.5 (4)	1.1 (2)
**Gram-positive**	*Enterococcus* spp.	4.5 (5)	10.4 (7)	3 (8)	5.9 (11)
	*E. faecium*	9 (10)	13.4 (9)	10.9 (29)	6.5 (12)
	*E. faecalis*	---	4.5 (3)	1.1 (3)	3.8 (7)
	VRE	8 (9)	3 (2)	7.5 (20)	10.8 (20)
	*Staphylococcus* spp.	7.1 (8)	4.5 (3)	7.5 (25)	4.3 (8)
	MRSA	0.9 (1)	3 (2)	---	---
	*Streptococcus* spp.	1.8 (2)	1.5 (1)	3.4 (9)	4.3 (8)
	*Clostridium* spp.	0.9 (1)	1.5 (1)	2.6 (7)	1.6 (3)
	Others	46 (83/179)		15 (67/451)	

Data are given as relative frequencies (%) and absolute numbers (*n*). Lead pathogens were considered the primarily causative pathogen identified. Overall, no pathogen was detected in 8.7% (36/414) of cases (subgroups: IB 10.5% (13/124), PI: 7.9% (23/290)) * Combined resistance towards 3rd-generation cephalosporins & fluoroquinolones ^#^ Combined resistances towards piperacillin/tazobactam, ceftazidime, fluoroquinolones & carbapenems.

**Table 3 antibiotics-10-00687-t003:** Primary endpoints in baseline cohort and propensity-score matched patients.

Baseline (*n* = 414) Propensity-Score Matched (*n* = 224)								
Intermittent Bolus		Prolonged Infusion	d%	*p*	Intermittent Bolus	Prolonged Infusion	d%	*p*
Hospital mortality, % (*n*)	64.5 (80/124)	48.3 (140/290)	**16.2**	**0.0024**	62.5 (70/112)	50.9 (57/112)	**11.6**	n.s.
ICU mortality, % (*n*)	57.3 (71/124)	43.5 (126/290)	**13.8**	**0.01**	54.5 (61/112)	44.6 (50/112)	**9.9**	n.s.
30-day mortality % (*n*)	54.8 (68/124)	41 (119/290)	**13.8**	**0.0097**	52.7 (59/112)	42 (47/112)	**10.7**	n.s.
90-day mortality % (*n*)	62.9 (78/124)	47.9 (139/290)	**15**	**0.005**	60.7 (68/112)	50.9 (57/112)	**9.8**	n.s.

Data are %(*n*) unless other specified. IB: intermittent bolus. PI: prolonged infusion. d%: percentage difference. ICU: Intensive Care Unit. n.s.: not significant.

## Data Availability

Data are made available on reasonable request through the principal investigators.

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
