# Peer review of "Prolonged Infusion of β-Lactams Decreases Mortality in Patients with Septic Shock: A Retrospective before-and-after Study"

_antibiotics, 2021, doi:10.3390/antibiotics10060687_

Round 1

Reviewer 1 Report

From the viewpoint of pharmacokinetics, it is not surprised to see prolonged infusion is better than intermittent bolus injection. 

This manuscript can be accepted after minor amendment.

(1) The sample size is small but uneven.

(2) Did the authors see prolonged infusion display display special benefits in specific pathogens? 

Author Response

Thank you very much for critically reviewing our manuscript. We appreciate the contribution. Below you will find detailed answers to your questions.

From the viewpoint of pharmacokinetics, it is not surprised to see prolonged infusion is better than intermittent bolus injection. 

The authors did not claim that clinical benefits of a PI of ß-lactams was a novel finding. It is rather the subgroup of patients we investigated (septic shock and abdominal focus). Patients in abdominal sepsis face an especially severe form of septic shock with massive cytokine release and pathophysiological changes that seriously alter PK of antibiotics. From that point of view, the authors think that the presented data add yet another piece of evidence that PI should be implemented into clinical routine and is save even within the sickest of patients we face in the ICU.

This manuscript can be accepted after minor amendment.

(1) The sample size is small but uneven.

Thank you for the comment. However, we think a sample size of 414 septic shock patients from a single center is quite remarkable. Most of the studies included far less patients and a broad variety of infectious foci. The sample size is unequal, indeed. The reason is that we wanted to minimize the influence of organizational changes. Between 2015-2020 the head of the ICU and the senior surgeon responsible for the decision of surgical interventions and the surgical procedure remained unchanged. After propensity-score matching, sample sizes are, however, equal (n=112 IB vs n=112 PI) and still big enough to draw conclusions and perform a solid statistical analysis.

(2) Did the authors see prolonged infusion display special benefits in specific pathogens? 

That is a fair question. We did not perform such an analysis as the focus of the manuscript lies on PK and clinical outcome. As can be seen from table 2, there is no alarming number of MDR pathogens. Investigation of the effect on specific pathogens, MIC distribution of these pathogens would be necessary which we cannot provide within 10 days from now.

Reviewer 2 Report

The introduction is well written and summarizes correctly the current knowledge in this field.

My major comments are:

The number of patients in the PI period is far above the number in the BI period, suggesting that some organizational modifications occurred during the period of the study. The modification of the percentage of Piperacillin-Tazobactam use seems also to indicate some organizational changes that might influence the duration of mechanical ventilation and other results of the study.

The study was conducted in a surgical ICU and infections were mainly abdominal. The surgical treatment of abdominal infections is clearly at least as important as the role of antibiotic therapy, and any change in the surgical team or the technique used can influence the results. Can you indicate if such changes occurred during this period, which could explain the evolution of the number of patients and influence the results?

The propensity-score matching process doesn’t include organ transplantation. I believe that immune suppression could clearly be included in the process of case/control choice of patients.

Since you have more than the double number of patients in the PI period, I believe that if possible, you can match 2 patients in PI period for each patient in the BI period, improving the statistical power of your analysis. If the number of surgeons is limited, at best, you should adjust the matching to the surgeon who operated the patient.

Furthermore, urinary tract infections do not generally pose any problem in terms of the concentration of antibiotics at the source of infection and I really believe that these patients should either be excluded from the study or alternatively be excluded in a sensitivity propensity matched analysis excluding urinary tract infections.

I believe that the fact that inflammation markers such as CRP or procalcitonin have the same kinetics in the 2 groups is clearly against a better infection control using PI of antibiotics. The authors should discuss this problem in the discussion chapter.

Minor comments.

Abstract line 33. The sentence indicating that ‘’After propensity-score matching, 30- and 90-day mortality remained lower for the PI group (-10%)’’ is in my opinion incorrect because the percentage is not significant. At least, the authors should indicate that the mortality was not significantly lower or indicate the P value.

Abstract Line 34 and 35. Given the retrospective nature of the study, you cannot say that the prolonged infusion reduced the duration of invasive ventilation or led to a stronger decrease in SOFA scores. You can just say that PI was associated with shorter durations of ventilation or a more rapid decrease in SOFA score, but the causal relationship cannot be established.

Line 37. You cannot say that PI reduces mortality. In a more careful sentence, you just can say that the period of PI might be associated with a reduction in mortality rate.

Materials & Methods

Page 2 Line 103: can you indicate why the period of March 2015 to July 2020 was chosen because you don’t have exactly the same duration for BI and PI periods.

Page 6, Table 2 : please correct  Fluoroquinolones

Page 7 , table 3 : please provide the P values instead of N.S.

Page 10 line 292. Also, you cannot say that your data suggest a significant reduction of mortality because this is not the case after propensity matching.

Author Response

Thank you very much for critically reviewing our manuscript. We appreciate the contribution. Below you will find detailed answers to your questions.

The number of patients in the PI period is far above the number in the BI period, suggesting that some organizational modifications occurred during the period of the study. The modification of the percentage of Piperacillin-Tazobactam use seems also to indicate some organizational changes that might influence the duration of mechanical ventilation and other.

Thank you for this comment. We discussed this finding while preparing the manuscript. Admittedly, the groups are unequal (IB n=125, PI n=290). But contrary to your conception, we specifically chose the period as there were no major organizational changes. Between 2015-2020 the head of anesthesiology and intensive care as well as the senior surgeon (responsible for surgical interventions, surgical procedures) remained unchanged. Additionally, leaving the SIRS-based definition of sepsis, more patients were diagnosed with sepsis and septic shock retrospectively. As discussed in the manuscript, we included patients meeting the effective SSC guidelines.

Regarding your comment concerning PIP/TAZ: The authors acknowledged possible explanations in the discussion. We mainly identified two factors for the increased use of PIP/TAZ in the PI group. 1) We updated the internal sepsis pathway in accordance with the SSC guideline revision. Together with microbiology and the ID-department (after revision of the pathogen distribution and resistance patterns) we encouraged the use of PIP/TAZ over carbapenems even in critically ill patients as the antibiotic of choice. 2) The patients included into the study were treated in our ICU when ESBL-phenotypes were still treated with PIP/TAZ (before the MERINO-study of Harris et al.) and EUCAST introduced “intermediate” sensitivity as a new category. As we cumulate patients at risk for MRE infections and ESBL-phenotypes in particular, these factors may explain the use of PIP/TAZ.

The study was conducted in a surgical ICU and infections were mainly abdominal. The surgical treatment of abdominal infections is clearly at least as important as the role of antibiotic therapy, and any change in the surgical team or the technique used can influence the results. Can you indicate if such changes occurred during this period, which could explain the evolution of the number of patients and influence the results?

You are absolutely right here. As mentioned before, the principle surgical decisions and major (surgical) treatment steps were advised by the same senior surgeon during the complete observation period. Evaluating outcome according to specific surgeons is, with all respect, not acceptable. Surgical source control does not solely depend on the individual surgeon’s skills and is often hampered due to circumstances that cannot be reflected in a statistical analysis. Specific data on surgical techniques or differences in surgical approaches cannot be delivered. It is not possible to obtain these data in a retrospective analysis. A prospective study with questionnaires to evaluate the surgical procedure would be necessary.

The propensity-score matching process doesn’t include organ transplantation. I believe that immune suppression could clearly be included in the process of case/control choice of patients.

SOT patients are a bit of a problem. On the one hand, as you indicated, this population of patients are interesting to look at because of the immunosuppressive therapy and consecutively severe infections. On the other hand, inclusion of SOT patients could hamper the generalization of the data as most of the ICUs do not deal with SOT on a regular basis. We opted against including these patients into the matching process mainly for two reasons: 1) SOT-groups did not significantly differ (SOT IB vs SOT PI p=0.49) and 2) sample size of the matched analysis: adding SOT as an additional parameter to the matching procedure would have decreased sample size further and jeopardize statistical validity. However, we are currently planning on publishing data on the SOT-subgroup including more patients.

Since you have more than the double number of patients in the PI period, I believe that if possible, you can match 2 patients in PI period for each patient in the BI period, improving the statistical power of your analysis. If the number of surgeons is limited, at best, you should adjust the matching to the surgeon who operated the patient.

We discussed your amendment with our statistician. While a comparison in a 2-to-1 manner is possible and might increase power, this approach on the other hand reduces sample remarkably. As mentioned above, surgical procedures were performed by senior surgeons. Analyzing the data according to the actual surgeon responsible on that specific day is counterproductive and opens the field to an irreproducible number of (possible) other factors (day – or nighttime-surgery, years of surgical experience, anesthetic management, hemodynamic management, …) that would make a reasonable analysis impossible.

Furthermore, urinary tract infections do not generally pose any problem in terms of the concentration of antibiotics at the source of infection and I really believe that these patients should either be excluded from the study or alternatively be excluded in a sensitivity propensity matched analysis excluding urinary tract infections.

The authors do not fully agree with you here. Of course, hydrophilic antibiotics with a predominantly renal clearance are unproblematic in UTIs. However, patients in septic shock with a high risk for AKI do not necessarily exhibit high serum concentrations and high concentrations at the site of infection. Patients in septic shock might have both: low serum concentrations due to an expansion of the volume of distribution and low concentrations at the site of infection due to renal impairment, failure and RRT. Furthermore, sepsis is a distinct disease (not just a severe form of infection). We find it illegitimate to exclude these patients just because we think that this focus is not much of a problem.

I believe that the fact that inflammation markers such as CRP or procalcitonin have the same kinetics in the 2 groups is clearly against a better infection control using PI of antibiotics. The authors should discuss this problem in the discussion chapter.

We addressed this problem in the discussion section.

Minor comments.

Abstract line 33. The sentence indicating that ‘’After propensity-score matching, 30- and 90-day mortality remained lower for the PI group (-10%)’’ is in my opinion incorrect because the percentage is not significant. At least, the authors should indicate that the mortality was not significantly lower or indicate the P value.

We inserted the indication that the finding was not statistically significant.

Abstract Line 34 and 35. Given the retrospective nature of the study, you cannot say that the prolonged infusion reduced the duration of invasive ventilation or led to a stronger decrease in SOFA scores. You can just say that PI was associated with shorter durations of ventilation or a more rapid decrease in SOFA score, but the causal relationship cannot be established.

Changed. Thank you.

Line 37. You cannot say that PI reduces mortality. In a more careful sentence, you just can say that the period of PI might be associated with a reduction in mortality rate.

Rephrased as amended.

Materials & Methods

Page 2 Line 103: can you indicate why the period of March 2015 to July 2020 was chosen because you don’t have exactly the same duration for BI and PI periods.

See explanation above (no changes in organizational structure)

Page 6, Table 2: please correct  Fluoroquinolones

Corrected

Page 7, table 3 : please provide the P values instead of N.S.

P-values are now embedded into table 3.

Page 10 line 292. Also, you cannot say that your data suggest a significant reduction of mortality because this is not the case after propensity matching.

By saying that our data “suggest” something, the authors do not claim to have “proven” causality and we thoroughly discussed the limitations of the retrospective nature of the study. We find that a 10%-reduction of i.e., ICU mortality is absolutely relevant to clinicians - regardless of the p-value. Propensity-score matching loses patients in the matching process and decreases group size. This alone may (or may not, of course) explain that the p-value does not attain significance level.

We rephrased the conclusion, but we still emphasize the clinical relevance of our findings.

Round 2

Reviewer 2 Report

I have the answers to my questions.